# LEARNING-BASED FREQUENCY ESTIMATION ALGORITHMS

**Chen-Yu Hsu, Piotr Indyk, Dina Katabi & Ali Vakilian**
Computer Science and Artificial Intelligence Lab
Massachusetts Institute of Technology
Cambridge, MA 02139, USA
{cyhsu,indyk,dk,vakilian}@mit.edu

## ABSTRACT

Estimating the frequencies of elements in a data stream is a fundamental task in data analysis and machine learning. The problem is typically addressed using streaming algorithms which can process very large data using limited storage. Today's streaming algorithms, however, cannot exploit patterns in their input to improve performance. We propose a new class of algorithms that automatically learn relevant patterns in the input data and use them to improve its frequency estimates. The proposed algorithms combine the benefits of machine learning with the formal guarantees available through algorithm theory. We prove that our learning-based algorithms have lower estimation errors than their non-learning counterparts. We also evaluate our algorithms on two real-world datasets and demonstrate empirically their performance gains.

## 1. INTRODUCTION

Classical algorithms provide formal guarantees over their performance, but often fail to leverage useful patterns in their input data to improve their output. On the other hand, deep learning models are highly successful at capturing and utilizing complex data patterns, but often lack formal error bounds. The last few years have witnessed a growing effort to bridge this gap and introduce algorithms that can adapt to data properties while delivering worst case guarantees. Deep learning modules have been integrated into the design of Bloom filters (Kraska et al., 2018; Mitzenmacher, 2018), caching algorithms (Lykouris & Vassilvitskii, 2018), graph optimization (Dai et al., 2017), similarity search (Salakhutdinov & Hinton, 2009; Weiss et al., 2009) and compressive sensing (Bora et al., 2017). This paper makes a significant step toward this vision by introducing frequency estimation streaming algorithms that automatically learn to leverage the properties of the input data.

Estimating the frequencies of elements in a data stream is one of the most fundamental subroutines in data analysis. It has applications in many areas of machine learning, including feature selection (Aghazadeh et al., 2018), ranking (Dzogang et al., 2015), semi-supervised learning (Talukdar & Cohen, 2014) and natural language processing (Goyal et al., 2012). It has been also used for network measurements (Estan & Varghese, 2003; Yu et al., 2013; Liu et al., 2016) and security (Schechter et al., 2010). Frequency estimation algorithms have been implemented in popular data processing libraries, such as Algebird at Twitter (Boykin et al., 2016). They can answer practical questions like: what are the most searched words on the Internet? or how much traffic is sent between any two machines in a network?

The frequency estimation problem is formalized as follows: given a sequence $S$ of elements from some universe $U$, for any element $i \in U$, estimate $f_i$, the number of times $i$ occurs in $S$. If one could store all arrivals from the stream $S$, one could sort the elements and compute their frequencies. However, in big data applications, the stream is too large (and may be infinite) and cannot be stored. This challenge has motivated the development of *streaming algorithms*, which read the elements of $S$ in a single pass and compute a good estimate of the frequencies using a limited amount of space.[1] Over the last two decades, many such streaming algorithms have been developed, including

---

[1] Specifically, the goal of the problem is as follows. Given a sequence $S$ of elements from $U$, the desired algorithm reads $S$ in a single pass while writing into memory $C$ (whose size can be much smaller than the length of $S$). Then, given any element $i \in U$, the algorithm reports an estimation of $f_i$ based only on the content of $C$.

Count-Sketch (Charikar et al., 2002), Count-Min (Cormode & Muthukrishnan, 2005b) and multi-stage filters (Estan & Varghese, 2003). The performance guarantees of these algorithms are well-understood, with upper and lower bounds matching up to $O(\cdot)$ factors (Jowhari et al., 2011).

However, such streaming algorithms typically assume generic data and do not leverage useful patterns or properties of their input. For example, in text data, the word frequency is known to be inversely correlated with the length of the word. Analogously, in network data, certain applications tend to generate more traffic than others. If such properties can be harnessed, one could design frequency estimation algorithms that are much more efficient than the existing ones. Yet, it is important to do so in a general framework that can harness various useful properties, instead of using handcrafted methods specific to a particular pattern or structure (e.g., word length, application type).

In this paper, we introduce learning-based frequency estimation streaming algorithms. Our algorithms are equipped with a learning model that enables them to exploit data properties without being specific to a particular pattern or knowing the useful property a priori. We further provide theoretical analysis of the guarantees associated with such learning-based algorithms.

We focus on the important class of "hashing-based" algorithms, which includes some of the most used algorithms such as Count-Min, Count-Median and Count-Sketch. Informally, these algorithms hash data items into $B$ buckets, count the number of items hashed into each bucket, and use the bucket value as an estimate of item frequency. The process can be repeated using multiple hash functions to improve accuracy. Hashing-based algorithms have several useful properties. In particular, they can handle item deletions, which are implemented by decrementing the respective counters. Furthermore, some of them (notably Count-Min) never underestimate the true frequencies, i.e., $\tilde{f}_i \geq f_i$ holds always. However, hashing algorithms lead to estimation errors due to collisions: when two elements are mapped to the same bucket, they affect each others' estimates. Although collisions are unavoidable given the space constraints, the overall error significantly depends on the pattern of collisions. For example, collisions between high-frequency elements ("heavy hitters") result in a large estimation error, and ideally should be minimized. The existing algorithms, however, use random hash functions, which means that collisions are controlled only probabilistically.

Our idea is to use a small subset of $S$, call it $S'$, to learn the heavy hitters. We can then assign heavy hitters their own buckets to avoid the more costly collisions. It is important to emphasize that we are learning the *properties* that identify heavy hitters as opposed to the *identities* of the heavy hitters themselves. For example, in the word frequency case, shorter words tend to be more popular. The subset $S'$ itself may miss many of the popular words, but whichever words popular in $S'$ are likely to be short. Our objective is *not* to learn the identity of high frequency words using $S'$. Rather, we hope that a learning model trained on $S'$ learns that short words are more frequent, so that it can identify popular words even if they did not appear in $S'$.

Our main contributions are as follows:

- We introduce *learning-based* frequency estimation streaming algorithms, which learn the properties of heavy hitters in their input and exploit this information to reduce errors

- We provide performance guarantees showing that our algorithms can deliver a logarithmic factor improvement in the error bound over their non-learning counterparts. Furthermore, we show that our learning-based instantiation of Count-Min, a widely used algorithm, is asymptotically optimal among *all* instantiations of that algorithm. See Table 4.1 in section 4.1 for the details.

- We evaluate our learning-based algorithms using two real-world datasets: traffic load on an Internet backbone link and search query popularity. In comparison to their non-learning counterparts, our algorithms yield performance gains that range from 18% to 71%.

## 2. RELATED WORK

**Frequency estimation in data streams.** Frequency estimation, and the closely related problem of finding frequent elements in a data stream, are some of the most fundamental and well-studied problems in streaming algorithms, see Cormode & Hadjieleftheriou (2008) for an overview. Hashing-based algorithms such as Count-Sketch (Charikar et al., 2002), Count-Min (Cormode & Muthukrishnan, 2005b) and multi-stage filters (Estan & Varghese, 2003) are widely used solutions for these problems. These algorithms also have close connections to sparse recovery and compressed sens-

ing (Candès et al., 2006; Donoho, 2006), where the hashing output can be considered as a compressed representation of the input data (Gilbert & Indyk, 2010).

Several "non-hashing" algorithms for frequency estimation have been also proposed (Misra & Gries, 1982; Demaine et al., 2002; Karp et al., 2003; Metwally et al., 2005). These algorithms do not possess many of the properties of hashing-based methods listed in the introduction (such as the ability to handle deletions), but they often have better accuracy/space tradeoffs. For a fair comparison, our evaluation focuses only on hashing algorithms. However, our approach for learning heavy hitters should be useful for non-hashing algorithms as well.

Some papers have proposed or analyzed frequency estimation algorithms customized to data that follows Zipf Law (Charikar et al., 2002; Cormode & Muthukrishnan, 2005a; Metwally et al., 2005; Minton & Price, 2014; Roy et al., 2016); the last algorithm is somewhat similar to the "lookup table" implementation of the heavy hitter oracle that we use as a baseline in our experiments. Those algorithms need to know the data distribution a priori, and apply only to one distribution. In contrast, our learning-based approach applies to any data property or distribution, and does not need to know that property or distribution a priori.

**Learning-based algorithms.** Recently, researchers have begun exploring the idea of integrating machine learning models into algorithm design. In particular, researchers have proposed improving compressed sensing algorithms, either by using neural networks to improve sparse recovery algorithms (Mousavi et al., 2017; Bora et al., 2017), or by designing linear measurements that are optimized for a particular class of vectors (Baldassarre et al., 2016; Mousavi et al., 2015), or both. The latter methods can be viewed as solving a problem similar to ours, as our goal is to design "measurements" of the frequency vector $(f_1, f_2 \ldots, f_{|U|})$ tailored to a particular class of vectors. However, the aforementioned methods need to explicitly represent a matrix of size $B \times |U|$, where $B$ is the number of buckets. Hence, they are unsuitable for streaming algorithms which, by definition, have space limitations much smaller than the input size.

Another class of problems that benefited from machine learning is *distance estimation*, i.e., compression of high-dimensional vectors into compact representations from which one can estimate distances between the original vectors. Early solutions to this problem, such as *Locality-Sensitive Hashing*, have been designed for worst case vectors. Over the last decade, numerous methods for learning such representations have been developed (Salakhutdinov & Hinton, 2009; Weiss et al., 2009; Jegou et al., 2011; Wang et al., 2016). Although the objective of those papers is similar to ours, their techniques are not usable in our applications, as they involve a different set of tools and solve different problems.

More broadly, there have been several recent papers that leverage machine learning to design more efficient algorithms. The authors of (Dai et al., 2017) show how to use reinforcement learning and graph embedding to design algorithms for graph optimization (e.g., TSP). Other learning-augmented combinatorial optimization problems are studied in (He et al., 2014; Balcan et al., 2018; Lykouris & Vassilvitskii, 2018). More recently, (Kraska et al., 2018; Mitzenmacher, 2018) have used machine learning to improve indexing data structures, including Bloom filters that (probabilistically) answer queries of the form "is a given element in the data set?" As in those papers, our algorithms use neural networks to learn certain properties of the input. However, we differ from those papers both in our design and theoretical analysis. Our algorithms are designed to reduce collisions between heavy items, as such collisions greatly increase errors. In contrast, in existence indices, all collisions count equally. This also leads to our theoretical analysis being very different from that in (Mitzenmacher, 2018).

## 3. PRELIMINARIES

### 3.1. ESTIMATION ERROR

We will use $e_i := |\tilde{f}_i - f_i|$ to denote the estimation error for $f_i$. To measure the overall estimation error between the frequencies $\mathcal{F} = \{f_1, f_2, \cdots, f_{|U|}\}$ and their estimates $\tilde{\mathcal{F}} = \{\tilde{f}_1, \tilde{f}_2, \cdots, \tilde{f}_{|U|}\}$, we will use the *expected* error $E_{i \sim \mathcal{D}}[e_i]$, where $\mathcal{D}$ models the distribution over the *queries* to the data structure. Similar to past work (Roy et al., 2016), we assume the query distribution $\mathcal{D}$ is the same as the distribution of the input stream, i.e., for any $j$ we have $\Pr_{i \sim \mathcal{D}}[i = j] = f_j/N$, where $N$

is the sum of all frequencies. This leads to the estimation error of $\tilde{\mathcal{F}}$ with respect to $\mathcal{F}$:

$$\mathrm{Err}(\mathcal{F}, \tilde{\mathcal{F}}) := 1/N \sum_{i \in U} |\tilde{f}_i - f_i| \cdot f_i \qquad (3.1)$$

We note that the theoretical guarantees of frequency estimation algorithms are typically phrased in the "$(\epsilon, \delta)$-form", e.g., $\Pr[|\tilde{f}_i - f_i| > \epsilon N] < \delta$ for *every* $i$ (see e.g., Cormode & Muthukrishnan (2005b)). However, this formulation involves two objectives ($\epsilon$ and $\delta$). We believe that the (single objective) expected error in Equation 3.1 is more natural from the machine learning perspective.

### 3.2. ALGORITHMS FOR FREQUENCY ESTIMATION

In this section, we recap three variants of hashing-based algorithms for frequency estimation.

**Single Hash Function.** The basic hashing algorithm uses a single uniformly random hash function $h : U \rightarrow [B]$, where we use $[B]$ to denote the set $\{1 \ldots B\}$. The algorithm maintains a one-dimensional array $C[1 \ldots B]$, in which all entries are initialized to 0. Given an element $i$, the algorithm increments $C[h(i)]$. It can be seen that at the end of the stream, we have $C[b] = \sum_{j:h(j)=b} f_j$. The estimate $\tilde{f}_i$ of $f_i$ is defined as $\tilde{f}_i = C[h(i)] = \sum_{j:h(i)=h(j)} f_j$. Note that it is always the case that $\tilde{f}_i \geq f_i$.

**Count-Min.** We have $k$ distinct hash functions $h_i : U \rightarrow [B]$ and an array $C$ of size $k \times B$. The algorithm maintains $C$, such that at the end of the stream we have $C[\ell, b] = \sum_{j:h_\ell(j)=b} f_j$. For each $i \in U$, the frequency estimate $\tilde{f}_i$ is equal to $\min_{\ell \leq k} C[\ell, h_\ell(i)]$, and always satisfies $\tilde{f}_i \geq f_i$.

**Count-Sketch.** Similarly to Count-Min, we have $k$ distinct hash functions $h_i : U \rightarrow [B]$ and an array $C$ of size $k \times B$. Additionally, in Count-Sketch, we have $k$ sign functions $g_i : U \rightarrow \{-1, 1\}$, and the algorithm maintains $C$ such that $C[\ell, b] = \sum_{j:h_\ell(j)=b} f_j \cdot g_\ell(j)$. For each $i \in U$, the frequency estimate $\tilde{f}_i$ is equal to the median of $\{g_\ell(i) \cdot C[\ell, h_\ell(i)]\}_{\ell \leq k}$. Note that unlike the previous two methods, here we may have $\tilde{f}_i < f_i$.

### 3.3. ZIPFIAN DISTRIBUTION

In our theoretical analysis we assume that the item frequencies follow the Zipf Law. That is, if we re-order the items so that their frequencies appear in a sorted order $f_{i_1} \geq f_{i_2} \geq \ldots \geq f_{i_n}$, then $f_{i_j} \propto 1/j$. To simplify the notation we assume that $f_i = 1/i$.

## 4. LEARNING-BASED FREQUENCY ESTIMATION ALGORITHMS

We aim to develop frequency estimation algorithms that exploit data properties for better performance. To do so, we learn an oracle that identifies heavy hitters, and use the oracle to assign each heavy hitter its unique bucket to avoid collisions. Other items are simply hashed using any classic frequency estimation algorithm (e.g., Count-Min, or Count-Sketch), as shown in the block-diagram in Figure 4.1. This design has two useful properties: First, it allows us to augment a classic frequency estimation algorithm with learning capabilities, producing a learning-based counterpart that inherits the original guarantees of the classic algorithm. For example, if the classic algorithm is Count-Min, the resulting learning-based algorithm never underestimates the frequencies. Second, it provably reduces the estimation errors, and for the case of Count-Min it is (asymptotically) optimal.

Algorithm 1 provides pseudo code for our design. The design assumes an oracle $\mathsf{HH}(i)$ that attempts to determine whether an item $i$ is a "heavy hitter" or not. All items classified as heavy hitters are assigned to one of the $B_r$ *unique* buckets reserved for heavy items. All other items are fed to the remaining $B - B_r$ buckets using a conventional frequency estimation algorithm $SketchAlg$ (e.g., Count-Min or Count-Sketch).

The estimation procedure is analogous. To compute $\tilde{f}_i$, the algorithm first checks whether $i$ is stored in a unique bucket, and if so, reports its count. Otherwise, it queries the $SketchAlg$ procedure. Note that if the element is stored in a unique bucket, its reported count is exact, i.e., $\tilde{f}_i = f_i$.

The oracle is constructed using machine learning and trained with a small subset of $S$, call it $S'$. Note that the oracle learns the *properties* that identify heavy hitters as opposed to the *identities* of the heavy hitters themselves. For example, in the case of word frequency, the oracle would learn that

shorter words are more frequent, so that it can identify popular words even if they did not appear in the training set $S'$.

## 4.1. ANALYSIS

Our algorithms combine simplicity with strong error bounds. Below, we summarize our theoretical results, and leave all theorems, lemmas, and proofs to the appendix. In particular, Table 4.1 lists the results proven in this paper, where each row refers to a specific streaming algorithm, its corresponding error bound, and the theorem/lemma that proves the bound.

First, we show (Theorem 9.11 and Theorem 9.14) that if the heavy hitter oracle is accurate, then the error of the learned variant of Count-Min is up to a logarithmic factor smaller than that of its non-learning counterpart. The improvement is maximized when $B$ is of the same order as $n$ (a common scenario[2]). Furthermore, we prove that this result continues to hold even if the learned oracle makes prediction errors with probability $\delta$, as long as $\delta = O(1/\ln n)$ (Lemma 9.15).

Second, we show that, asymptotically, our learned Count-Min algorithm cannot be improved any further by designing a better hashing scheme. Specifically, for the case of Learned Count-Min with a perfect oracle, our design achieves the same asymptotic error as the "Ideal Count-Min", which optimizes its hash function for the given input (Theorem 10.4).

Finally, we note that the learning-augmented algorithm inherits any $(\epsilon, \delta)$-guarantees of the original version. Specifically, its error is not larger than that of $SketchAlg$ with space $B - B_r$, for any input.

| Algorithms | Expected Error | Analysis |
|---|---|---|
| Single Hash Function ($B$) | $\Theta(\ln^2 n/B)$ | Lemma 9.2 |
| Count-Min Sketch ($k, B$) | $\Omega(\frac{k \ln n}{B \ln k})$ and $O(\frac{k \ln n \ln^{\frac{k+2}{k-1}}(\frac{kn}{B})}{B})$ | Theorem 9.11 |
| Learned Count-Min($B$) | $\Theta(\ln^2(n/B)/B)$ | Theorem 9.14 |
| Ideal Count-Min($B$) | $\Theta(\ln^2(n/B)/B)$ | Theorem 10.4 |

Table 4.1: Our performance bounds for different algorithms on streams with frequencies obeying Zipf Law. $k$ is a constant ($\geq 2$) that refers to the number of hash functions, $B$ is the number of buckets, and $n$ is the number of distinct elements. The space complexity of all algorithms is the same, $\Theta(B)$. See section 9.4 for non-asymptotic versions of the some of the above bounds

## 5. EXPERIMENTS

**Baselines.** We compare our learning-based algorithms with their non-learning counterparts. Specifically, we augment Count-Min with a learned oracle using Algorithm 1, and call the learning-augmented algorithm "Learned Count-Min". We then compare Learned Count-Min with traditional Count-Min. We also compare it with "Learned Count-Min with Ideal Oracle" where the neural-network oracle is replaced with an ideal oracle that knows the identities of the heavy hitters in the test data, and "Table Lookup with Count-Min" where the heavy hitter oracle is replaced with a lookup table that memorizes heavy hitters in the training set. The comparison with the latter baseline allows

[2]For example, Goyal et al. (2012) uses $B = 20M, n = 33.5M$ and $B = 50M, n = 94M$.

---

**Algorithm 1** Learning-Based Frequency Estimation

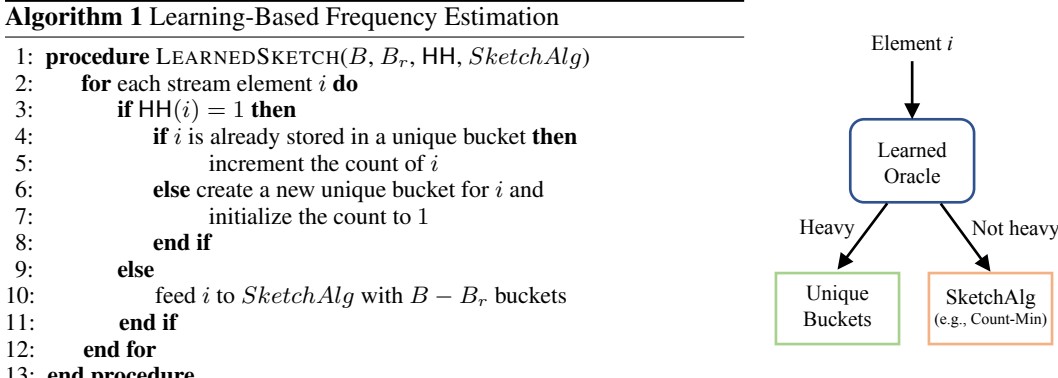

```
 1: procedure LEARNEDSKETCH(B, B_r, HH, SketchAlg)
 2:     for each stream element i do
 3:         if HH(i) = 1 then
 4:             if i is already stored in a unique bucket then
 5:                 increment the count of i
 6:             else create a new unique bucket for i and
 7:                 initialize the count to 1
 8:             end if
 9:         else
10:             feed i to SketchAlg with B − B_r buckets
11:         end if
12:     end for
13: end procedure
```

Figure 4.1: Pseudo-code and block-diagram representation of our algorithms

us to show the ability of Learned Count-Min to generalize and detect heavy items unseen in the training set. We repeat the evaluation where we replace Count-Min (CM) with Count-Sketch (CS) and the corresponding variants. We use validation data to select the best $k$ for all algorithms.

**Training a Heavy Hitter Oracle.** We construct the heavy hitter oracle by training a neural network to predict the heaviness of an item. Note that the prediction of the network is not the final estimation. It is used in Algorithm 1 to decide whether to assign an item to a unique bucket. We train the network to predict the item counts (or the log of the counts) and minimize the squared loss of the prediction. Empirically, we found that when the counts of heavy items are few orders of magnitude larger than the average counts (as is the case for the Internet traffic data set), predicting the log of the counts leads to more stable training and better results. Once the model is trained, we select the optimal cutoff threshold using validation data, and use the model as the oracle described in Algorithm 1.

### 5.1. INTERNET TRAFFIC ESTIMATION

For our first experiment, the goal is to estimate the number of packets for each network flow. A flow is a sequence of packets between two machines on the Internet. It is identified by the IP addresses of its source and destination and the application ports. Estimating the size of each flow $i$ – i.e., the number of its packets $f_i$ – is a basic task in network management (Sivaraman et al., 2017).

**Dataset**: The traffic data is collected at a backbone link of a Tier1 ISP between Chicago and Seattle in 2016 (CAIDA). Each recording session is around one hour. Within each minute, there are around 30 million packets and 1 million unique flows. For a recording session, we use the first 7 minutes for training, the following minute for validation, and estimate the packet counts in subsequent minutes. The distribution of packet counts over Internet flows is heavy tailed, as shown in Figure 5.1.

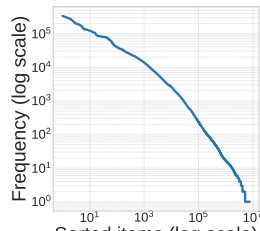

Figure 5.1: Frequency of Internet Flows

**Model**: The patterns of the Internet traffic are very dynamic, i.e., the flows with heavy traffic change frequently from one minute to the next. However, we hypothesize that the space of IP addresses should be smooth in terms of traffic load. For example, data centers at large companies and university campuses with many students tend to generate heavy traffic. Thus, though the individual flows from these sites change frequently, we could still discover regions of IP addresses with heavy traffic through a learning approach.

We trained a neural network to predict the log of the packet counts for each flow. The model takes as input the IP addresses and ports in each packet. We use two RNNs to encode the source and destination IP addresses separately. The RNN takes one bit of the IP address at each step, starting from the most significant bit. We use the final states of the RNN as the feature vector for an IP address. The reason to use RNN is that the patterns in the bits are hierarchical, i.e., the more significant bits govern larger regions in the IP space. Additionally, we use two-layer fully-connected networks to encode the source and destination ports. We then concatenate the encoded IP vectors, encoded port vectors, and the protocol type as the final features to predict the packet counts [3]. The inference time takes 2.8 microseconds per item on a single GPU without any optimizations[4].

**Results**: We plot the results of two representative test minutes (the 20th and 50th) in Figure 5.2. All plots in the figure refer to the estimation error (Equation 3.1) as a function of the used space. The space includes space for storing the buckets and the model. Since we use the same model for all test minutes, the model space is amortized over the 50-minute testing period.

Figure 5.2 reveals multiple findings. First, the figure shows that our learning-based algorithms exhibit a better performance than their non-learning counterparts. Specifically, Learned Count-Min, compared to Count-Min, reduces the the error by 32% with space of 0.5 MB and 42% with space of 1.0 MB (Figure 5.2a). Learned Count-Sketch, compared to Count-Sketch, reduces the error by 52%

---

[3]We use RNNs with 64 hidden units. The two-layer fully-connected networks for the ports have 16 and 8 hidden units. The final layer before the prediction has 32 hidden units.

[4]Note that new specialized hardware such as Google TPU, hardware accelerators and network compression (Han et al., 2017; Sze et al., 2017; Chen et al., 2017; Han et al., 2016; 2015) can drastically improve the inference time. Further, Nvidia has predicted that GPU will get 1000x faster by 2025. Because of these trends, the overhead of neural network inference is expected to be less significant in the future (Kraska et al., 2018).

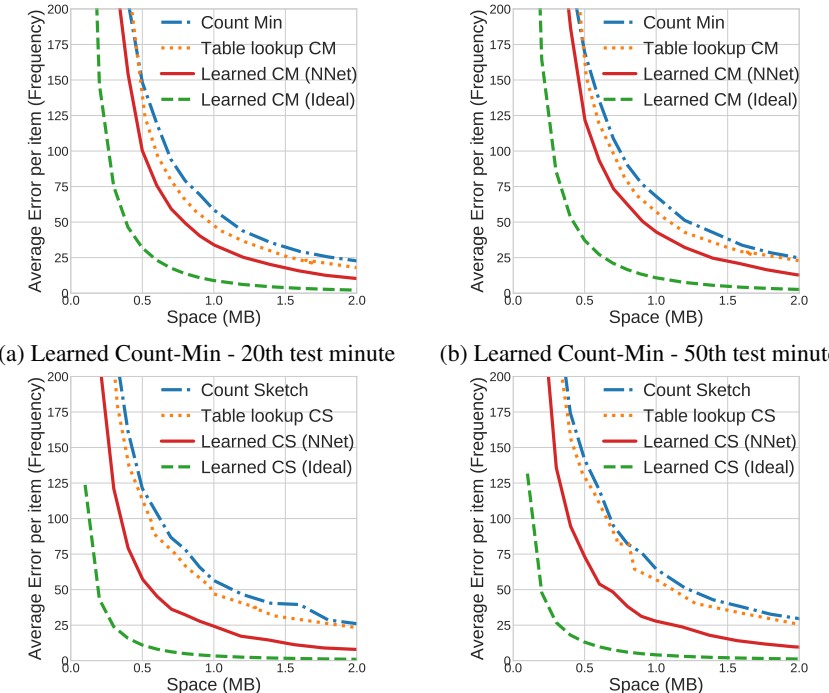

(a) Learned Count-Min - 20th test minute   (b) Learned Count-Min - 50th test minute

(c) Learned Count-Sketch - 20th test minute   (d) Learned Count-Sketch - 50th test minute

Figure 5.2: Comparison of our algorithms with Count-Min and Count-Sketch on Internet traffic data.

at 0.5 MB and 57% at 1.0 MB (Figure 5.2c). In our experiments, each regular bucket takes 4 bytes. For the learned versions, we account for the extra space needed for the unique buckets to store the item IDs and the counts. One unique bucket takes 8 bytes, twice the space of a normal bucket.[5]

Second, the figure also shows that our neural-network oracle performs better than memorizing the heavy hitters in a lookup table. This is likely due to the dynamic nature of Internet traffic –i.e., the heavy flows in the training set are significantly different from those in the test data. Hence, memorization does not work well. On the other hand, our model is able to extract structures in the input that generalize to unseen test data.

Third, the figure shows that our model's performance stays roughly the same from the 20th to the 50th minute (Figure 5.2b and Figure 5.2d), showing that it learns properties of the heavy items that generalize over time.

Lastly, although we achieve significant improvement over Count-Min and Count-Sketch, our scheme can potentially achieve even better results with an ideal oracle, as shown by the dashed green line in Figure 5.2. This indicates potential gains from further optimizing the neural network model.

### 5.2. SEARCH QUERY ESTIMATION

For our second experiment, the goal is to estimate the number of times a search query appears.

**Dataset**: We use the AOL query log dataset, which consists of 21 million search queries collected from 650 thousand users over 90 days. The users are anonymized in the dataset. There are 3.8 million unique queries. Each query is a search phrase with multiple words (e.g., "periodic table element poster"). We use the first 5 days for training, the following day for validation, and estimate the number of times different search queries appear in subsequent days. The distribution of search query frequency follows the Zipfian law, as shown in Figure 5.3

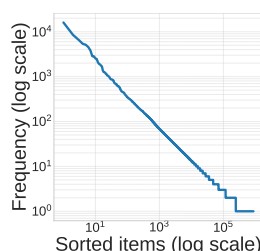

Figure 5.3: Frequency of search queries

---

[5]By using hashing with open addressing, it suffices to store IDs hashed into $\log B_r + t$ bits (instead of whole IDs) to ensure there is no collision with probability $1 - 2^{-t}$. $\log B_r + t$ is comparable to the number of bits per counter, so the space for a unique bucket is twice the space of a normal bucket.

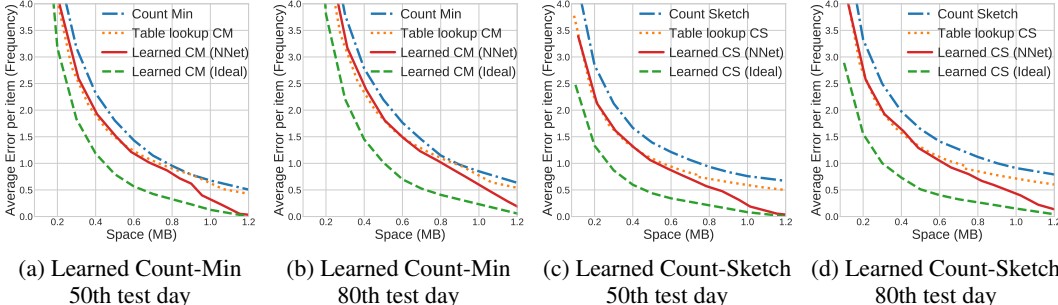

(a) Learned Count-Min
50th test day

(b) Learned Count-Min
80th test day

(c) Learned Count-Sketch
50th test day

(d) Learned Count-Sketch
80th test day

Figure 5.4: Comparison of our algorithms with Count-Min and Count-Sketch on search query data.

**Model**: Unlike traffic data, popular search queries tend to appear more consistently across multiple days. For example, "google" is the most popular search phrase in most of the days in the dataset. Simply storing the most popular words can easily construct a reasonable heavy hitter predictor. However, beyond remembering the popular words, other factors also contribute to the popularity of a search phrase that we can learn. For example, popular search phrases appearing in slightly different forms may be related to similar topics. Though not included in the AOL dataset, in general, metadata of a search query (e.g., the location of the search) can provide useful context of its popularity.

To construct the heavy hitter oracle, we trained a neural network to predict the number of times a search phrase appears. To process the search phrase, we train an RNN with LSTM cells that takes characters of a search phrase as input. The final states encoded by the RNN are fed to a fully-connected layer to predict the query frequency. Our character vocabulary includes lower-case English alphabets, numbers, punctuation marks, and a token for unknown characters. We map the character IDs to embedding vectors before feeding them to the RNN[6]. We choose RNN due to its effectiveness in processing sequence data (Sutskever et al., 2014; Graves, 2013; Kraska et al., 2018).

**Results**: We plot the estimation error vs. space for two representative test days (the 50th and 80th day) in Figure 5.4. As before, the space includes both the bucket space and the space used by the model. The model space is amortized over the test days since the same model is used for all days.

Similarly, our learned sketches outperforms their conventional counterparts. For Learned Count-Min, compared to Count-Min, it reduces the loss by 18% at 0.5 MB and 52% at 1.0 MB (Figure 5.4a). For Learned Count-Sketch, compared to Count-Sketch, it reduces the loss by 24% at 0.5 MB and 71% at 1.0 MB (Figure 5.4c). Further, our algorithm performs similarly for the 50th and the 80th day (Figure 5.4b and Figure 5.4d), showing that the properties it learns generalize over a long period.

The figures also show an interesting difference from the Internet traffic data: memorizing the heavy hitters in a lookup table is quite effective in the low space region. This is likely because the search queries are less dynamic compared to Internet traffic (i.e., top queries in the training set are also popular on later days). However, as the algorithm is allowed more space, memorization becomes ineffective.

## 5.3. ANALYZING HEAVY HITTER MODELS

We analyze the accuracy of the neural network heavy hitter models to better understand the results on the two datasets. Specifically, we use the models to predict whether an item is a heavy hitter (top 1% in counts) or not, and plot the ROC curves in Figure 5.5. The figures show that the model for the Internet traffic data has learned to predict heavy items more effectively, with an AUC score of 0.9. As for the model for search query data, the AUC score is 0.8. This also explains why we see larger improvements over non-learning algorithms in Figure 5.2.

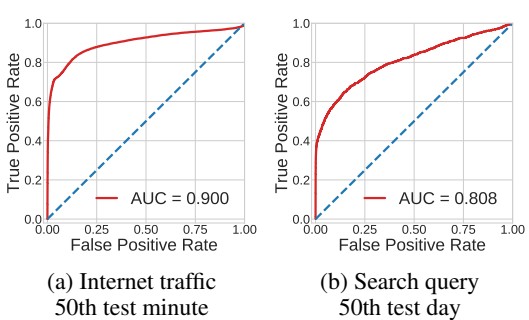

(a) Internet traffic
50th test minute

(b) Search query
50th test day

Figure 5.5: ROC curves of the learned models.

---

[6]We use an embedding size of 64 dimensions, an RNN with 256 hidden units, and a fully-connected layer with 32 hidden units.

## 6. EMBEDDING SPACE VISUALIZATION

In this section, we visualize the embedding spaces learned by our heavy hitter models to shed light on the properties or structures the models learned. Specifically, we take the neural network activations before the final fully-connected layer, and visualize them in a 2-dimensional space using t-SNE (Maaten & Hinton, 2008). To illustrate the differences between heavy hitters (top 1% in counts) and the rest ("light" items), we randomly sample an equal amount of examples from both classes. We visualize the embedding space for both the Internet traffic and search query datasets.

We show the embedding space learned by the model on the Internet traffic data in Figure 6.1. Each point in the scatter plot represents one Internet traffic flow. By coloring each flow with its number of packets in Figure 6.1a, we see that the model separate flows with more packets (green and yellow clusters) from flows with fewer packets (blue clusters). To understand what structure the model learns to separate these flows, we color each flow with its destination IP address in Figure 6.1b. We found that clusters with more packets are often formed by flows sharing similar destination address prefixes. Interestingly, the model learns to group flows with similar IP prefixes closer in the embedding space. For example, the dark blue cluster at the upper left of Figure 6.1b shares a destination IP address prefix "1.96.*.*". Learning this "structure" from the Internet traffic data allows the model to generalize to packets unseen in the training set.

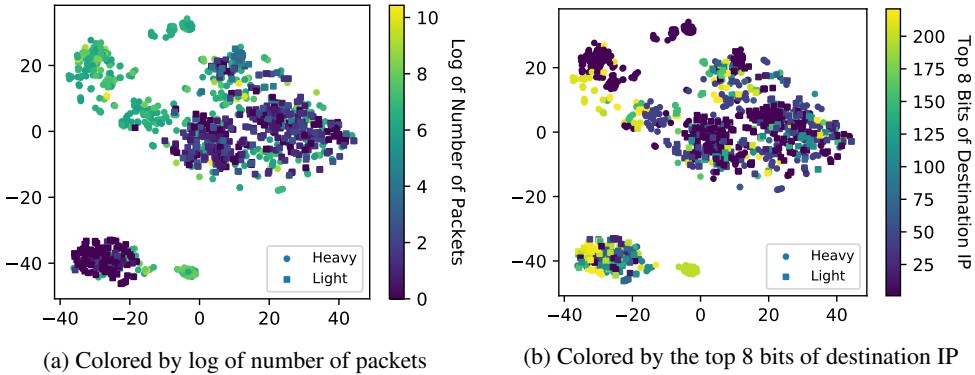

(a) Colored by log of number of packets

(b) Colored by the top 8 bits of destination IP

Figure 6.1: Visualization of the embedding space learned by our model on the Internet traffic data. Each point in the scatter plot represents one network flow.

We show the embedding space learned by the model on the search query data in Figure 6.2. Each point in the scatter plot represents one search query. Similarly, the model learns to separate frequent search queries from the rest in Figure 6.2a. By coloring the queries with the number of characters in Figure 6.2b, we have multiple interesting findings. First, queries with similar length are closer in the embedding space, and the y-axis forms the dimension representing query length. Second, if we simply use the query length to predict heavy hitters, many light queries will be misclassified. The model must have learned other structures to separate heavy hitters in Figure 6.2a.

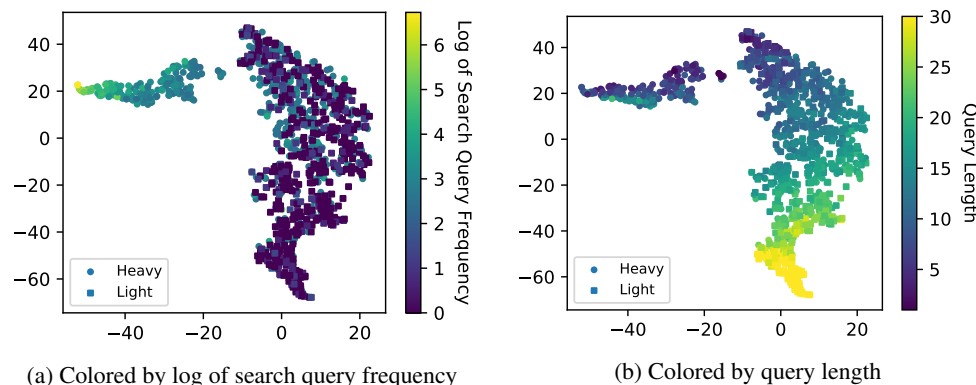

(a) Colored by log of search query frequency

(b) Colored by query length

Figure 6.2: Visualization of the embedding space learned by our model on the search query data. Each point in the scatter plot represents one search query.

## 7. Conclusion

We have presented a new approach for designing frequency estimation streaming algorithms by augmenting them with a learning model that exploits data properties. We have demonstrated the benefits of our design both analytically and empirically. We envision that our work will motivate a deeper integration of learning in algorithm design, leading to more efficient algorithms.

## 8. Acknowledgements

The authors would like to thank the anonymous reviewers for helpful comments. The work was partially supported by NSF TRIPODS award #1740751, NSF AITF award #1535851 and Simons Investigator Award. We also thank the various companies sponsoring the MIT Center for Wireless Networks and Mobile Computing.

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

## 9. ANALYSIS - FULL PROOFS

In this section, we analyze the performance of three different approaches, single (uniformly random) hash function, Count-Min sketch, and Learned Count-Min sketch when the frequency of items is from Zipfian distribution. For simplicity, we assume that the number of distinct elements $n$ is equal to the size of the universe $|U|$, and $f_i = 1/i$. We use $[n]$ to denote the set $\{1 \ldots n\}$. We also drop the normalization factor $1/N$ in the definition of estimation error.

The following observation is useful throughout this section (in particular, in the section on *non-asymptotic analysis*).

**Observation 9.1.** *For sufficiently large values of* $n$ *(i.e.,* $n > 250$*),*

$$\ln(n+1) < \sum_{i=1}^{n} \frac{1}{i} < \ln n + 0.58 \tag{9.1}$$

### 9.1. SINGLE HASH FUNCTION

**Lemma 9.2.** *The expected error of a single uniformly random hash function* $h : [n] \to [B]$ *(with* $B$ *buckets) for estimating the frequency of items whose distribution is Zipfian is* $\Theta(\frac{\ln^2 n}{B})$.

*Proof:*

$$\mathbf{E}[\mathrm{Err}(\mathcal{F}, \tilde{\mathcal{F}}_h)] = \mathbf{E}[\sum_{j \in [n]} (\tilde{f}_j - f_j) \cdot f_j] = \sum_{j \in [n]} \mathbf{E}[\sum_{\{q \neq j \mid h[q] = h[j]\}} f_q] \cdot f_j$$

$$= \sum_{j \in [n]} (\Theta(\frac{\ln n}{B}) - \frac{f_j}{B}) \cdot f_j = \Theta(\frac{\ln^2 n}{B}). \qquad \square$$

Moreover, since each bucket maintains the frequency of items that are mapped to it under $h$, the space complexity of this approach is proportional to the number of buckets which is $\Theta(B)$.

### 9.2. COUNT-MIN SKETCH

Here, we provide an upper bound and lower bound for the expected estimation error of Count-Min sketch with $k$ hash functions and $B$ buckets per row. In the rest of this section, for each $j \in [n], \ell \leq k$, we use $e_{j,\ell}$ and $e_j$ respectively to denote the estimation error of $f_j$ by $h_\ell$ and Count-Min sketch. Recall that the expected error of Count-Min sketch is defined as follows:

$$\mathbf{E}[\mathrm{Err}(\mathcal{F}, \tilde{\mathcal{F}}_{CM})] = \mathbf{E}[\sum_{j \in [n]} (\tilde{f}_j - f_j) \cdot f_j] = \sum_{j \in [n]} \mathbf{E}[e_j] \cdot f_j, \tag{9.2}$$

Our high-level approach is to partition the interval $[0, B \ln n]$ into $m + 1$ smaller intervals by a sequence of thresholds $\Theta(\ln^{1+\gamma}(\frac{n}{B})) = r_0 \leq \cdots \leq r_m = B \ln n$ where $\gamma$ is a parameter to be determined later. Formally, we define the sequence of $r_i$s to satisfy the following property:

$$\forall \ell > 0, \quad r_\ell := (\ln(\frac{n}{B}) + \ln r_{\ell+1}) \ln^\gamma r_{\ell+1}. \tag{9.3}$$

**Claim 9.3.** *For each* $i \geq 0$, $\ln r_{i+1} \geq 2 \ln r_i$.

*Proof:* By (9.3) and assuming $\ln r_{i+1} \geq \ln(\frac{n}{B})$, $r_i < 2 \ln^{1+\gamma} r_{i+1}$. Hence, $\ln r_{i+1} > (\frac{r_i}{2})^{\frac{1}{1+\gamma}} > 2 \ln r_i$ for sufficiently large values of $r_i$[7] assuming $\gamma \leq 3$.

Note that as long as $\ln r_{i+1} \geq \ln(\frac{n}{B})$, $r_i \leq r_{i+1}$. Otherwise, $r_i = o(\ln^{1+\gamma}(\frac{n}{B})) < r_0$. $\qquad \square$

**Corollary 9.4.** *For each* $i \geq 1$ *and* $c \geq 1$, $\sum_{j \geq i} \ln^{-c} r_j = O(\ln^{-c} r_i)$.

---

[7]More precisely, $r_i \geq 1.3 \times 10^6$.

Then, to compute (9.2), we rewrite $\mathbf{E}[e_j]$ using the thresholds $r_0, \cdots, r_m$ as follows:

$$\mathbf{E}[e_j] = \int_0^\infty \Pr(e_j \geq x) dx \tag{9.4}$$

$$= \int_0^{\frac{r_0}{B}} \Pr(e_j \geq x) dx + \sum_{i=0}^{m-1} \int_{\frac{r_i}{B}}^{\frac{r_{i+1}}{B}} \Pr(e_j \geq x) dx$$

$$\leq O(\frac{r_0}{B}) + \sum_{i=0}^{m-1} \int_{\frac{r_i}{B}}^{\frac{r_{i+1}}{B}} \left( \Pr(e_j \geq x | e_j < \frac{r_{i+1}}{B}) + \sum_{q=i+1}^{m-1} \Pr(e_j \geq \frac{r_q}{B} | e_j < \frac{r_{q+1}}{B}) \right) dx$$

Next, we compute $\Pr(e_j \geq \frac{t}{B} | e_j < \frac{r_{i+1}}{B})$ for each $t \in [r_i, r_{i+1})$.

**Lemma 9.5.** *For each $t \in [r_i, r_{i+1})$, $\Pr(e_j \geq \frac{t}{B} | e_j < \frac{r_{i+1}}{B}) = O\left( \frac{k(\ln(\frac{n}{B}) + \ln r_{i+1})}{t \ln^\gamma r_{i+1}} \right)$.*

*Proof:* First we prove the following useful observation.

**Claim 9.6.** *For each item $j$ and hash function $h_\ell$, $\mathbf{E}[e_{j,\ell} \mid e_{j,\ell} < \frac{r}{B}] \leq \frac{\ln(\frac{n}{B}) + \ln r}{B}$.*

*Proof:* Note that the condition $e_{j,\ell} < \frac{r}{B}$ implies that for all items $q < \frac{B}{r}$ (and $q \neq j$), $h_\ell(j) \neq h_\ell(q)$. Hence,

$$\mathbf{E}[e_{j,\ell} \mid e_{j,\ell} < \frac{r}{B}] \leq \sum_{q > \frac{B}{r}} f_q \cdot \frac{1}{B} \leq \frac{\ln n - \ln(\frac{B}{r})}{B} = \frac{\ln(\frac{n}{B}) + \ln r}{B}. \qquad \square$$

Thus, by Markov's inequality, for each item $j$ and hash function $h_\ell$,

$$\Pr(e_{j,\ell} > \frac{t}{B} \mid e_{j,\ell} \leq \frac{r}{B}) \leq \frac{\ln(\frac{n}{B}) + \ln r}{t}. \tag{9.5}$$

Now, for each $h_\ell$ in Count-Min sketch, we bound the value of $\Pr(e_{j,\ell} \geq \frac{t}{B})$ where $t \in [r_i, r_{i+1})$:

$$\Pr(e_{j,\ell} \geq \frac{t}{B}) \leq \sum_{q=i}^{m-1} \Pr(e_{j,\ell} \geq \frac{r_q}{B} | e_{j,\ell} < \frac{r_{q+1}}{B})$$

$$\leq \sum_{q=i}^{m-1} \frac{\ln(\frac{n}{B}) + \ln r_{q+1}}{r_q} = O(\frac{1}{\ln^\gamma r_{i+1}}) \rhd \text{ by (9.5) and Corollary 9.4} \tag{9.6}$$

Hence, for $k \geq 2$,

$$\Pr(e_j \geq \frac{t}{B} | e_j < \frac{r_{i+1}}{B}) \leq k \Pr(e_{j,1} \geq \frac{t}{B} | e_{j,1} < \frac{r_{i+1}}{B}) \Pi_{\ell=2}^k \Pr(e_{j,\ell} \geq \frac{t}{B})$$

$$= O\left( \frac{k(\ln(\frac{n}{B}) + \ln r_{i+1})}{t \ln^{\gamma(k-1)} r_{i+1}} \right) \rhd \text{ by (9.5) and (9.6)} \qquad \square$$

Next, for each item $j$, we bound the contribution of each interval $(\frac{r_i}{B}, \frac{r_{i+1}}{B})$ to $\mathbf{E}[e_j]$, namely $\int_{\frac{r_i}{B}}^{\frac{r_{i+1}}{B}} \Pr(e_j \geq x) dx$.

**Claim 9.7.** *For each $i \geq 0$, $\int_{\frac{r_i}{B}}^{\frac{r_{i+1}}{B}} \Pr(e_j \geq x) dx = O(\frac{k(\ln(\frac{n}{B}) + \ln r_{i+1})}{B \ln^{\gamma(k-1)-1} r_{i+1}})$.*

*Proof:*

$$\int_{\frac{r_i}{B}}^{\frac{r_{i+1}}{B}} \Pr(e_j \geq x) dx = \int_{\frac{r_i}{B}}^{\frac{r_{i+1}}{B}} \left( \underbrace{\Pr(e_j \geq x | e_j < \frac{r_{i+1}}{B})}_{A} + \sum_{q=i+1}^{m-1} \underbrace{\Pr(e_j \geq \frac{r_q}{B} | e_j < \frac{r_{q+1}}{B})}_{B_q} \right) dx$$

First, we compute $\int_{\frac{r_i}{B}}^{\frac{r_{i+1}}{B}} A dx$.

$$\int_{\frac{r_i}{B}}^{\frac{r_{i+1}}{B}} \Pr(e_j \geq x | e_j < \frac{r_{i+1}}{B}) dx \leq \int_{\frac{r_i}{B}}^{\frac{r_{i+1}}{B}} O(\frac{k(\ln(\frac{n}{B}) + \ln r_{i+1})}{(B \ln^{\gamma(k-1)} r_{i+1}) x}) dx$$

$$= O(\frac{k(\ln(\frac{n}{B}) + \ln r_{i+1})}{B \ln^{\gamma(k-1)-1} r_{i+1}}) \tag{9.7}$$

Similarly, $\int_{\frac{r_i}{B}}^{\frac{r_{i+1}}{B}} \sum_{q=i+1}^{m-1} B_q dx$ is at most:

$$\int_{\frac{r_i}{B}}^{\frac{r_{i+1}}{B}} \sum_{q=i+1}^{m-1} \Pr(e_j \geq \frac{r_q}{B}|e_j < \frac{r_{q+1}}{B})dx = \int_{\frac{r_i}{B}}^{\frac{r_{i+1}}{B}} \sum_{q=i+1}^{m-1} O(\frac{k(\ln(\frac{n}{B}) + \ln r_{q+1})}{r_q \ln^{\gamma(k-1)} r_{q+1}})dx$$

$$= \int_{\frac{r_i}{B}}^{\frac{r_{i+1}}{B}} \sum_{q=i+1}^{m-1} O(\frac{k}{\ln^{\gamma k} r_{q+1}})dx \quad \triangleright \text{ by (9.3)}$$

$$= \int_{\frac{r_i}{B}}^{\frac{r_{i+1}}{B}} O(\frac{k}{\ln^{\gamma k} r_{i+2}})dx \quad \triangleright \text{ by Corollary 9.4}$$

$$= O(\frac{kr_{i+1}}{B \ln^{\gamma k} r_{i+2}}) = O(\frac{k(\ln(\frac{n}{B}) + \ln r_{i+2})}{B \ln^{\gamma(k-1)} r_{i+2}}) \quad (9.8)$$

Hence,

$$\int_{\frac{r_i}{B}}^{\frac{r_{i+1}}{B}} \Pr(e_j \geq x)dx = \int_{\frac{r_i}{B}}^{\frac{r_{i+1}}{B}} \left( \Pr(e_j \geq x|e_j < \frac{r_{i+1}}{B}) + \sum_{q=i+1}^{m-1} \Pr(e_j \geq \frac{r_q}{B}|e_j < \frac{r_{q+1}}{B}) \right)dx$$

$$= O(\frac{k(\ln(\frac{n}{B}) + \ln r_{i+1})}{B \ln^{\gamma(k-1)-1} r_{i+1}}) + O(\frac{k(\ln(\frac{n}{B}) + \ln r_{i+2})}{B \ln^{\gamma(k-1)} r_{i+2}}) \quad \triangleright \text{ by (9.7)-(9.8)}$$

$$= O(\frac{k(\ln(\frac{n}{B}) + \ln r_{i+1})}{B \ln^{\gamma(k-1)-1} r_{i+1}}) \quad \triangleright \text{ by (9.3)} \quad (9.9)$$

$\square$

Now, we complete the error analysis of (9.4):

$$\mathbf{E}[e_j] \leq O(\frac{r_0}{B}) + \sum_{i=0}^{m-1} \int_{\frac{r_i}{B}}^{\frac{r_{i+1}}{B}} \left( \Pr(e_j \geq x|e_j < \frac{r_{i+1}}{B}) + \sum_{q=i+1}^{m-1} \Pr(e_j \geq \frac{r_q}{B}|e_j < \frac{r_{q+1}}{B}) \right)dx$$

$$= O(\frac{r_0}{B}) + \sum_{i=0}^{m-1} O(\frac{k(\ln(\frac{n}{B}) + \ln r_{i+1})}{B \ln^{\gamma(k-1)-1} r_{i+1}}) \quad \triangleright \text{ by (9.9)}$$

$$= O(\frac{r_0}{B}) + O(k\frac{\ln(\frac{n}{B})}{B \ln^{\gamma(k-1)-1} r_1}) + O(\frac{k}{B \ln^{\gamma(k-1)-2} r_1}) \quad (9.10)$$

Note that (9.10) requires $\gamma(k-1) - 2 \geq 1$ which is satisfied by setting $\gamma = 3/(k-1)$ and $k \geq 2$. Thus, for each item $j$,

$$\mathbf{E}[e_j] = O(r_0/B) = O(\frac{\ln^{\frac{k+2}{k-1}}(\frac{n}{B})}{B}) \quad (9.11)$$

**Lemma 9.8.** *The expected error of Count-Min sketch of size $k \times B$ (with $k \geq 2$) for estimating items whose frequency distribution is Zipfian is $O(\frac{\ln n \ln^{\frac{k+2}{k-1}}(n/B)}{B})$. In particular, if $B = \Theta(n)$, then $\mathbf{E}[\mathrm{Err}(\mathcal{F}, \tilde{\mathcal{F}}_{CM})] = O(\frac{\ln n}{n})$.*

*Proof:* By plugging in our upper bound on the estimation error of each item computed in (9.11) in the definition of expected estimation error of Count-Min (9.2), we have the following.

$$\mathbf{E}[\mathrm{Err}(\mathcal{F}, \tilde{\mathcal{F}}_{CM})] = \sum_{j=1}^{n} \mathbf{E}[e_j] \cdot f_j = O(\frac{\ln^{\frac{k+2}{k-1}}(\frac{n}{B})}{B} \cdot \ln n). \quad \square$$

Next, we show a lower bound on the expected error of Count-Min sketch with $B$ buckets (more precisely, of size $(k \times B/k)$) for estimating the frequency of items that follow Zipf Law.

**Observation 9.9.** *For each item $j$, $\Pr[e_j \geq 1/(2(\frac{B}{k}) \ln k + 1)] \geq 1 - \frac{1}{k}$.*

*Proof:* For each item $j$, the probability that none of the first $2(\frac{B}{k}) \ln k + 1$ items (excluding itself) collide with it in at least one of the hash functions $h_1, \cdots, h_k$ is at most $k(1 - \frac{1}{(B/k)})^{2(\frac{B}{k}) \ln k} \leq \frac{1}{k}$.

Hence, with probability at least $1 - \frac{1}{k}$, the estimation $\tilde{f}_j$ includes the frequency of one of the top $2(\frac{B}{k}) \ln k + 1$ frequent items. $\qquad \square$

**Lemma 9.10.** *The expected error of Count-Min sketch of size $k \times (\frac{B}{k})$ for estimating items whose frequency distribution is Zipfian is at least $\frac{(k-1) \ln n}{2B \ln k + k}$.*

*Proof:* Observation 9.9 implies that $\mathbf{E}[e_j] \geq (1 - \frac{1}{k}) \frac{1}{2(\frac{B}{k}) \ln k + 1} = \frac{k-1}{2B \ln k + k}$. Hence,

$$\mathbf{E}[\mathrm{Err}(\mathcal{F}, \tilde{\mathcal{F}}_{CM})] = \sum_{j \in [n]} \mathbf{E}[e_j] \cdot f_j$$

$$\geq (\frac{k-1}{2B \ln k + k}) \ln n \quad \triangleright \text{By Observation 9.1} \qquad \square$$

**Theorem 9.11.** *The expected error of Count-Min sketch of size $k \times (\frac{B}{k})$ on estimating the frequency of $n$ items from Zipfian distribution is at least $\Omega(\frac{k \ln n}{B \ln k})$ and at most $O(\frac{k \ln n \ln^{\frac{k+2}{k-1}}(\frac{kn}{B})}{B})$.*

*In particular, for the case $B = \Theta(n)$ and $k = O(1)$, the expected error of Count-Min sketch is $\Theta(\frac{\ln n}{B})$.*

*Proof:* The proof follows from Lemma 9.8 and 9.10. We remark that the bound in Lemma 9.8 is for the expected estimation error of Count-Min sketch of size $k \times B$. Hence, to get the bound on the expected error of Count-Min of size $k \times (\frac{B}{k})$, we must replace $B$ with $B/k$. $\qquad \square$

### 9.3. LEARNED COUNT-MIN SKETCH

**Definition 9.12 ($\phi$-HeavyHitter).** *Given a set of items $\mathcal{I} = \{i_1, \cdots, i_n\}$ with frequencies $\vec{f} = \langle f_1, \cdots, f_n \rangle$, an item $j$ is a $\phi$-HeavyHitter of $\mathcal{I}$ if $f_j \geq \phi ||\vec{f}||_1$.*

**Remark 9.13.** *If the frequency distribution of items $\mathcal{I}$ is Zipfian, then the number of $\phi$-HeavyHitters is at most $1/(\phi \ln n)$. In other words, $B_r \leq (\phi \ln n)^{-1}$.*

To recall, in our Learned Count-Min sketch with parameters $(B_r, B)$, $B_r$ buckets are reserved for the frequent items returned by HH and the rest of items are fed to a Count-Min sketch of size $k \times (\frac{B - B_r}{k})$ where $k$ is a parameter to be determined. We emphasize that the space complexity of Learned Count-Min sketch with parameter $(B_r, B)$ is $B_r + B = O(B)$.

**Theorem 9.14.** *The optimal expected error of Learned Count-Min sketches with parameters $(B_r, B)$ is at most $\frac{(\ln(n/B_r) + 0.58)^2}{B - B_r}$. In particular, for $B_r = \Theta(B) = \Theta(n)$, $\mathbf{E}[\mathrm{Err}(\mathcal{F}, \tilde{\mathcal{F}}_{L-CM})] = O(\frac{1}{n})$.*

*Proof:* Since, the count of top $B_r$ frequent items are stored in their own buckets, for each $j \leq B_r$, $e_j = 0$. Hence,

$$\mathbf{E}[\mathrm{Err}(\mathcal{F}, \tilde{\mathcal{F}}_{L-CM})] = \sum_{j \in [n]} \mathbf{E}[e_j] \cdot f_j$$

$$= \sum_{j > B_r} \mathbf{E}[e_j] \cdot f_j \quad \triangleright \forall j \leq B_r, e_j = 0$$

$$< \frac{(\ln(n/B_r) + 0.58)^2}{B - B_r} \quad \triangleright \text{By Observation 9.1}$$

Note that the last inequality follows from the guarantee of single hash functions; in other words, setting $k = 1$ in the Count-Min sketch. $\qquad \square$

#### 9.3.1. LEARNED COUNT-MIN SKETCH USING NOISY HEAVYHITTERS ORACLE

Unlike the previous part, here we assume that we are given a *noisy* HeavyHitters oracle $\mathsf{HH}_\delta$ such that for each item $j$, $\Pr\left(\mathsf{HH}_\delta(j, \frac{1}{B_r \ln n}) \neq \mathsf{HH}_0(j, \frac{1}{B_r \ln n})\right) \leq \delta$ where $\mathsf{HH}_0$ is an ideal HeavyHitter oracle that detects heavy items with no error.

**Lemma 9.15.** *In an optimal Learned Count-Min sketch with parameters $(B_r, B)$ and a noisy HeavyHitters oracle $\mathsf{HH}_\delta$, $\mathbf{E}[\mathrm{Err}(\mathcal{F}, \tilde{\mathcal{F}}_{(L-CM,\delta)})] = O(\frac{\delta^2 \ln^2 B_r + \ln^2(n/B_r)}{B - B_r})$.*

*Proof:* The key observation is that each heavy item, any of $B_r$ most frequent items, may only misclassify with probability $\delta$. Hence, for each item $j$ classified as "*not heavy*",

$$\mathbf{E}[e_j] \leq \delta \cdot \left(\frac{\ln B_r + 1}{B - B_r}\right) + \left(\frac{\ln(n/B_r) + 1}{B - B_r}\right) = O\left(\frac{\delta \ln B_r + \ln(n/B_r)}{B - B_r}\right), \quad (9.12)$$

where the first term denotes the expected contribution of the misclassified heavy items and the second term denotes the expected contribution of non-heavy items.

The rest of analysis is similar to the proof of Theorem 9.14.

$$\begin{aligned}
\mathbf{E}[\mathrm{Err}(\mathcal{F}, \tilde{\mathcal{F}}_{(L-CM,\delta)})] &= \sum_{j \in [n]} \mathbf{E}[e_j] \cdot f_j \\
&< \delta \sum_{j \leq B_r} \mathbf{E}[e_j] \cdot f_j + \sum_{j > B_r} \mathbf{E}[e_j] \cdot f_j \\
&\leq O(\delta \cdot \ln B_r + \ln(n/B_r)) \cdot O\left(\frac{\delta \ln B_r + \ln(n/B_r)}{B - B_r}\right) \quad \triangleright \text{ By (9.12)} \\
&\leq O\left(\frac{\delta^2 \ln^2 B_r + \ln^2(n/B_r)}{B - B_r}\right). \qquad\qquad\qquad\qquad \square
\end{aligned}$$

**Corollary 9.16.** *Assuming $B_r = \Theta(B) = \Theta(n)$ and $\delta = O(1/\ln n)$, $\mathbf{E}[\mathrm{Err}(\mathcal{F}, \tilde{\mathcal{F}}_{(L-CM,\delta)})] = O(1/n)$.*

**Space analysis.** Here, we compute the amount of space that is required by this approach.

**Lemma 9.17.** *The amount of space used by Learned Count-Min sketch of size $k \cdot \left(\frac{B - B_r}{k}\right)$ with cutoff ($B_r$ reserved buckets) for estimating the frequency of items whose distribution is Zipfian is $O(B)$.*

*Proof:* The amount of space required to store the counters corresponding to functions $h_1, \cdots, h_k$ is $O(k \cdot (\frac{B - B_r}{k}))$. Here, we also need to keep a mapping from the heavy items (top $B_r$ frequent items according to $\mathsf{HH}_\delta$) to the reserved buckets $B_r$ which requires extra $O(B_r)$ space; each reserved buckets stores both the hashmap of its corresponding item and its count. $\qquad\square$

### 9.4. Non-asymptotic Analysis of Count-Min and Learned Count-Min

In this section, we compare the non-asymptotic expected error of Count-Min sketch and out Learned Count-Min sketch with ideal HeavyHitters oracle. Throughout this section, we assume that the amount of available space to the frequency estimation algorithms is $(1 + \alpha)B$ words. More precisely, we compare the expected error of Count-Min sketch with $k$ hash functions and our Learned Count-Min sketch with $B_r = \alpha B$ reserved buckets. Recall that we computed the following bounds on the expected error of these approaches (Lemma 9.10 and Theorem 9.14):

$$\mathbf{E}[\mathrm{Err}_{CM}] \geq \frac{(k-1)\ln n}{2(1+\alpha)B \ln(k) + k}, \quad \mathbf{E}[\mathrm{Err}_{L-CM}] \leq \frac{(\ln(\frac{n}{\alpha B}) + 0.58)^2}{(1-\alpha)B}.$$

In the rest of this section, we assume that $B \geq \gamma n$ and then compute the minimum value of $\gamma$ that guarantees $\mathbf{E}[\mathrm{Err}_{L-CM}] \leq \frac{1}{1+\varepsilon} \cdot \mathbf{E}[\mathrm{Err}_{CM}]$. In other words, we compute the minimum amount of space that is required so that our Learned Count-Min sketch performs better than Count-Min sketch by a factor of at least $(1 + \varepsilon)$.

$$\mathbf{E}[\mathrm{Err}_{L-CM}] \leq \frac{(\ln(\frac{1}{\alpha\gamma}) + 0.58)^2}{(1-\alpha)B} \leq \frac{1}{1+\varepsilon} \cdot \frac{(k-1)\ln n}{2(1+\alpha)B \ln k + k} \leq \frac{1}{1+\varepsilon} \cdot \mathbf{E}[\mathrm{Err}_{CM}],$$

Hence, we must have $(0.58 - \ln \alpha - \ln \gamma)^2 \leq \frac{(k-1)(1-\alpha)}{(1+\varepsilon)(2(1+\alpha)\ln k + (\frac{k}{B}))} \cdot \ln n$. By solving the corresponding quadratic equation,

$$\ln^2 \gamma + 2(\ln \alpha - 0.58)\ln \gamma + \left((\ln \alpha - 0.58)^2 - \frac{(k-1)(1-\alpha)}{(1+\varepsilon)(2(1+\alpha)\ln k + (\frac{k}{B}))} \cdot \ln n\right) \leq 0.$$

This implies that $\ln \gamma = -\ln \alpha + 0.58 - \sqrt{\frac{(k-1)(1-\alpha)}{(1+\varepsilon)(2(1+\alpha)\ln k + (\frac{k}{B}))} \cdot \ln n}$. By setting $\alpha = \frac{1}{2}$, $\gamma \leq$

$\frac{3.58}{e^{\sqrt{\frac{(k-1)\ln n}{2(1+\varepsilon)(3\ln k + (k/B))}}}}$.

Next, we consider different values of $k$ and show that in each case what is the minimum amount of space in which Learned CM outperforms CM by a factor of $1.06$ (setting $\varepsilon = 0.06$).

- **$k = 1$**. In this case, we are basically comparing the expected error of a single hash function and Learned Count-Min. In particular, in order to get a gap of at least $(1 + \varepsilon)$, by a more careful analysis of Lemma 9.2, $\gamma$ must satisfy the following condition:
$$\mathbf{E}[\mathrm{Err}_{L-CM}] \leq \frac{(\ln(\frac{1}{\alpha\gamma}) + 0.58)^2}{(1 - \alpha)B} \leq \frac{1}{1 + \varepsilon} \cdot \frac{\ln^2 n - 1.65}{(1 + \alpha)B} \leq \frac{1}{1 + \varepsilon} \cdot \mathbf{E}[\mathrm{Err}_h],$$
To simplify it further, we require that $(\ln(\frac{2}{\gamma}) + 0.58)^2 \leq 3.18 \cdot (\ln^2 n - 1.65)$ which implies that $\gamma = \Theta(1/\ln n)$.

- **$k = 2$**. In this case, $\gamma \leq \frac{3.58}{e^{\sqrt{(\ln n)/4.5}}}$ for sufficiently large values of $B$. Hence, we require that the total amount of available space is at least $\frac{5.37n}{e^{\sqrt{(\ln n)/4.5}}}$.

- **$k \in \{3, 4\}$**: In this case, $\gamma \leq \frac{2}{e^{\sqrt{(\ln n)/3.5}}}$ for sufficiently large values of $B$. Hence, we require that the total amount of available space is at least $\frac{5.37n}{e^{\sqrt{(\ln n)/3.5}}}$.

- **$k \geq 5$**. In this case, $\gamma \leq \frac{2}{e^{\sqrt{(\ln n)/2.6}}}$ for sufficiently large values of $B$. Hence, we require that the total amount of available space is at least $\frac{5.37n}{e^{\sqrt{(\ln n)/2.6}}}$.

We also note that settings where the number of buckets is close to $n$ are quite common in practice. For example, Goyal et al. (2012) uses $((1+\alpha)B = 20M, n = 33.5M)$ and $((1+\alpha)B = 50M, n = 94M)$ with $k = 3$, while Kumar et al. (2004) uses different pairs of $((1 + \alpha)B, n)$ including $(B = 288K, n = 563K)$. Plugging these values into the exact formulas computed above shows that Learned CM has a significantly lower loss than CM.

## 10. OPTIMAL HASH FUNCTION

In this section we compute an asymptotic bound error bound of optimal hash functions that map items $[n]$ to a set of buckets $C[1 \cdots B]$.

**Claim 10.1.** *For a set of items $I$, let $f(I)$ denote the total frequency of all items in $I$; $f(I) := \sum_{i \in I} f_i$. In any optimal hash function of form $\{h : I \to [B_I]\}$ with minimum estimation error, any item with frequency at least $\frac{f(I)}{B_I}$ does not collide with any other items in $I$.*

*Proof:* For each bucket $b \in [B_I]$, lets $f_h(b)$ denotes the frequency of items mapped to $b$ under $h$; $f(b) := \sum_{i \in I : h(i) = b} f_i$. Recall that the estimation error of a hash function $h$ is defined as $\mathrm{Err}(\mathcal{F}(I), \tilde{\mathcal{F}}_h(I)) := \sum_{i \in I} f_i \cdot (f(h(i)) - i)$. Note that we can rewrite $\mathrm{Err}(\mathcal{F}(I), \tilde{\mathcal{F}}_h(I))$ as
$$\mathrm{Err}(\mathcal{F}(I), \tilde{\mathcal{F}}_h(I)) = \sum_{i \in I} f_i \cdot (f(h(i)) - f_i) = \sum_{i \in I} f_i \cdot f(h(i)) - \sum_{i \in I} f_i^2$$
$$= \sum_{b \in [B_I]} f(b)^2 - \sum_{i \in I} f_i^2. \tag{10.1}$$
Note that in (10.1) the second term is independent of $h$ and is a constant. Hence, an optimal hash function minimizes the first term, $\sum_{b \in B_I} f(b)^2$.

Suppose that an item $i^*$ with frequency at least $\frac{f(I)}{B_I}$ collides with a (non-empty) set of items $I^* \subseteq I \setminus \{i^*\}$ under an optimal hash function $h^*$. Since the total frequency of the items mapped to the bucket $b^*$ containing $i^*$ is greater than $\frac{f(I)}{B_I}$ (i.e., $f(h(i^*)) > \frac{f(I)}{B_I}$), there exists a bucket $\bar{b}$ such that $f(\bar{b}) < \frac{f(I)}{B_I}$. Next, we define a new hash function $\bar{h}$ with smaller estimation error compared to $h^*$ which contradicts the optimality of $h^*$:
$$\bar{h}(i) = \begin{cases} h^*(i) & \text{if } i \in I \setminus I^* \\ \bar{b} & \text{otherwise.} \end{cases}$$

Formally,
$$
\begin{aligned}
\mathrm{Err}(\mathcal{F}(I), \tilde{\mathcal{F}}_{h^*}(I)) - \mathrm{Err}(\mathcal{F}(I), \tilde{\mathcal{F}}_{\bar{h}}(I)) &= f_{h^*}(b^*)^2 + f_{h^*}(\bar{b})^2 - f_{\bar{h}}(b^*)^2 - f_{\bar{h}}(\bar{b})^2 \\
&= (f_{i^*} + f(I^*))^2 + f_{h^*}(\bar{b})^2 - f_{i^*}^2 - (f_{h^*}(\bar{b}) + f(I^*))^2 \\
&= 2 f_{i^*} \cdot f(I^*) - 2 f_{h^*}(\bar{b}) \cdot f(I^*) \\
&= 2 f(I^*) \cdot (f_{i^*} - f_{h^*}(\bar{b})) \\
&> 0 \quad \triangleright \text{ Since } f_{i^*} > \frac{f(I)}{B} > f_{h^*}(\bar{b}). \qquad \square
\end{aligned}
$$

Next, we show that in any optimal hash function $h^* : [n] \to [B]$ and assuming Zipfian input distribution, $\Theta(B)$ most frequent items do not collide with any other items under $h^*$.

**Lemma 10.2.** *Suppose that $B = n/\gamma$ where $\gamma \geq e^{4.2}$ is a constant and lets assume that items follow Zipfian distribution. In any hash function $h^* : [n] \to [B]$ with minimum estimation error, none of the $\frac{B}{2\ln\gamma}$ most frequent items collide with any other items (i.e., they are mapped to a singleton bucket).*

*Proof:* Let $i_{j^*}$ be the most frequent item that is not mapped to a singleton bucket under $h^*$. If $j^* > \frac{B}{2\ln\gamma}$ then the statement holds. Suppose it is not the case and $j^* \leq \frac{B}{2\ln\gamma}$. Let $I$ denote the set of items with frequency at most $f_{j^*} = 1/j^*$ (i.e., $I = \{i_j \mid j \geq j^*\}$) and let $B_I$ denote the number of buckets that the items with index at least $j^*$ mapped to; $B_I = B - j^* + 1$. Also note that by Observation 9.1, $f(I) < \ln(\frac{n}{j^*}) + 1$. Next, by Claim 10.1, we show that $h^*$ does not hash the items $\{j^*, \cdots, n\}$ to $B_I$ optimally. In particular, we show that the frequency of item $j^*$ is more than $\frac{f(I)}{B_I}$. To prove this, first we observe that the function $g(j) := j \cdot (\ln(n/j) + 1)$ is strictly increasing in $[1, n]$. Hence, for any $j^* \leq \frac{B}{2\ln\gamma}$,

$$
\begin{aligned}
j^* \cdot (\ln(\frac{n}{j^*}) + 1) &\leq \frac{B}{2\ln\gamma} \cdot (\ln(2\gamma\ln\gamma) + 1) \\
&\leq B \cdot (1 - \frac{1}{2\ln\gamma}) \quad \triangleright \text{ Since } \ln(2\ln\gamma) + 2 < \ln\gamma \text{ for } \ln\gamma \geq 4.2 \\
&< B_I
\end{aligned}
$$

Thus, $f_{j^*} = \frac{1}{j^*} > \frac{\ln(n/j^*)+1}{B_I} > \frac{f(I)}{B_I}$ which implies that an optimal hash function must map $j^*$ to a singleton bucket. $\qquad \square$

**Theorem 10.3.** *If $n/B \geq e^{4.2}$, then the estimation error of any hash function that maps a set of $n$ items following Zipfian distribution to $B$ buckets is $\Omega(\frac{\ln^2(\frac{n}{B})}{B})$.*

*Proof:* By Lemma 10.2, in any hash function with minimum estimation error, the $(\frac{B}{2\ln\gamma})$ most frequent items do not collide with any other items (i.e., they are mapped into a singleton bucket) where $\gamma = n/B > e^{4.2}$.

Hence, the goal is to minimize (10.1) for the set of items $I$ which consist of all items other than the $(\frac{B}{2\ln\gamma})$ most frequent items. Since the sum of squares of $m$ items that summed to $S$ is at least $S^2/m$, the multi-set loss of any optimal hash function is at least:

$$
\begin{aligned}
\mathrm{Err}(\mathcal{F}(I), \tilde{\mathcal{F}}_{h^*}(I)) &= \sum_{b \in [B]} f(b)^2 - \sum_{i \in [n]} f_i^2 \quad &&\triangleright \text{ By (10.1)} \\
&\geq \frac{(\sum_{i \in I} f_i)^2}{B(1 - \frac{1}{2\ln\gamma})} - \sum_{i \in I} f_i^2 \\
&\geq \frac{(\ln(2\gamma\ln\gamma) - 1)^2}{B} - \frac{2\ln\gamma}{B} + \frac{1}{n} \quad &&\triangleright \text{ By Observation 9.1} \\
&= \Omega(\frac{\ln^2\gamma}{B}) \quad &&\triangleright \text{ Since } \gamma > e^4 \\
&= \Omega(\frac{\ln^2(n/B)}{B}). \quad &&\square
\end{aligned}
$$

Next, we show a more general statement which basically shows that the estimation error of any Count-Min sketch with $B$ buckets is $\Omega(\frac{\ln^2(n/B)}{B})$ no matter how many rows (i.e., the value of $k$) it has.

**Theorem 10.4.** *If $n/B \geq e^{4.2}$, then the estimation error of any Count-Min sketch that maps a set of $n$ items following Zipfian distribution to $B$ buckets is $\Omega(\frac{\ln^2(\frac{n}{B})}{B})$.*

*Proof:* We prove the statement by showing a reduction that given a Count-Min sketch $CM(k)$ with hash functions $h_1, \cdots, h_k \in \{h : [n] \to [B/k]\}$ constructs a single hash function $h^* : [n] \to [B]$ whose estimation error is less than or equal to the estimation error of $CM(k)$.

For each item $i$, we define $C'[i]$ to be the bucket whose value is returned by $CM(k)$ as the estimate of $f_i$; $C'[i] := \arg\min_{j \in [k]} C[j, h_j(i)]$. Since the total number of buckets in $CM(k)$ is $B$, $|\{C'[i] \mid i \in [n]\}| \leq B$; in other words, we only consider the subset of buckets that $CM(k)$ uses to report the estimates of $\{f_i \mid i \in [n]\}$ which trivially has size at most $B$. We define $h^*$ as follows:
$$h^*(i) = (j^*, h_{j^*}(i)) \quad \rhd \text{ for each } i \in [n], \text{ where } j^* = \arg\min_{j \in [k]} C[j, h_j(i)]$$
Unlike $CM(k)$, $h^*$ maps each item to exactly one bucket in $\{C[\ell, j] \mid \ell \in [k], j \in [B/k]\}$; hence, for each item $i$, $C'[h^*(i)] \leq C[h^*(i)] = \tilde{f}_i$ where $\tilde{f}_i$ is the estimate of $f_i$ returned by $CM(k)$. Moreover, since for each $i$, $C'[h^*(i)] \geq f_i$,
$$\text{Err}(\mathcal{F}, \tilde{\mathcal{F}}_{h^*}) \leq \text{Err}(\mathcal{F}, \tilde{\mathcal{F}}_{CM(k)}).$$
Finally, by Theorem 10.3, the estimation error of $h^*$ is $\Omega(\frac{\ln^2(n/B)}{B})$ which implies that the estimation error of $CM(k)$ is $\Omega(\frac{\ln^2(n/B)}{B})$ as well. $\qquad\square$

