# OpenReview forum: "Learning-Based Frequency Estimation Algorithms"
_ICLR.cc/2019/Conference_

### Official Review · AnonReviewer1 · 2018-11-06
**Interesting topic, somewhat trivial algorithms and somewhat narrow results**

**Rating:** 6
**Confidence:** 4

**Review:**

This paper introduces the study of the problem of frequency estimation algorithms with machine learning advice. The problem considered is the standard frequency estimation problem in data streams where the goal is to estimate the frequency of the i-th item up to an additive error, i.e. the |\tilde f_i - f_i| should be minimized where \tilde f_i is the estimate of the true frequency f_i.

Pros:
-- Interesting topic of using machine learned advice to speed up frequency estimation is considered
-- New rigorous bounds are given on the complexity of frequency estimation under Zipfian distribution using machine learned advice
-- Experiments are given to justify claimed improvements in performance

Cons:

-- While the overall claim of the paper in the introduction seems to be to speed up frequency estimation using machine learned advice, results are only given for the Zipfian distribution.

-- The overall error model in this paper, which is borrowed from Roy et al. is quite restrictive as at it assumes that the queries to the frequency estimation data structure are coming from the same distribution as that given by f_i’s themselves. While in some applications this might be natural, this is certainly very restrictive in situations where f_i’s are updated not just by +/-1 increments but through arbitrary +/-Delta updates, as in this case it might be more natural to assume that the distribution of the queries might be proportional to the frequency that the corresponding coordinate is being updated, for example.

-- The algorithm proposed in the paper is very straightforward and just removes heavy hitters using oracle advice and then hashes everything else using the standard CountMin sketch.

-- Since CounMin is closely related to Bloom filters the idea of using machine learning to speed it up appears to be noticeably less novel given that for Bloom filters this has already been done by Mitzenmacher’18.

-- The analysis is relatively straightforward and boils down to bucketing the error and integration over the buckets.


Other comments:
-- The machine learned advice is assumed to be flawless at identifying the Heavy Hitters, authors might want to consider incorporating errors in the analysis.

---

> ### Author Response · Authors · 2018-11-15
> **Response to Reviewer 1**
>
> Thank you for the thoughtful comments. We are glad that you found our topic interesting and appreciated our theoretical analysis and experimental results. We address other comments below:
>
> [Results are only given for the Zipfian distribution]
> Many real-world data naturally follow the Zipf’s Law, as we showed in Figure 5.1 and Figure 5.3 for internet traffic and search query data. Thus, our theoretical analysis assumes item frequencies follow the Zipfian distribution. While our analysis makes this assumption, our algorithm does not have any assumption on the frequency distribution.
>
> [Assuming query distribution is the same as data distribution]
> As the reviewer pointed out, the query distribution we use is a natural choice. There might be other types of query distributions, such as the one pointed out by the reviewer.  Intuitively, our overall approach that separates heavy hitters from the rest should still be beneficial to such query distribution.
>
> [Algorithm design]
> We agree that our algorithms are relatively simple. We believe this is a feature not a bug: as we showed in Sec. 4.1, our algorithm does not need to be more complex. Specifically, our Learned Count-Min algorithm achieves the same asymptotic error as the “Ideal Count-Min”, which is allowed to optimize the whole hash function for the specific given input (Theorem 7.14 and Theorem 8.4 in Table 4.1). The proof of this statement demonstrates that identifying heavy hitters and placing them in unique bins is an (asymptotically) optimal strategy. (In fact, our first attempt at solving the problem was a much more complex algorithm which optimized the allocation of elements to the buckets (i.e., the whole hash function h) to minimize the error. This turned out to be unnecessary, as per the above argument.)
>
> [Novelty compared to Mitzenmacher’ 18]
> Our paper, as well as the works of Kraska et al ’18, Mitzenmacher ’18,  Lykouris &
> Vassilvitskii ’18, Purohit et al, NIPS’18, belong to a growing class of studies that use a machine learning oracle to improve the performance of algorithms. All such papers use a learned oracle of some form. The key differences are in what the oracle does, how it is used, and what can be proved about it. In Kraska’18 and Mitzenmacher’18, the oracle tries to directly solve the main problem, which is: “is the element in the set?” An analogous approach in our case would be to train an oracle that directly outputs the frequency of each element. However, instead of trying to directly solve the main problem (estimate the frequency of each element), our oracle is a subroutine that tries to predict the best resource allocation --i.e., it tries to answer the question of which elements should be given their own buckets and which should share with others.
>
> There are other differences.  For example, the main goal of our algorithm is to reduce collisions between heavy items, as such collisions greatly increase errors. This motivates our design to split heavy and light items using the learned model, and apply separate algorithms for each type. In contrast, in existence indices, all collisions count equally.
>
> Finally, our theoretical analysis is different from M'18 due to the intrinsic differences between the two problems, as outlined in the previous paragraph.
>
> [The analysis is relatively straightforward]
> There are three main theorems in our paper: Theorem 8.4, Theorem 7.11 and 7.14.  Our proofs of Theorem 7.11 and 7.14 are technically involved, even if the techniques are relatively standard.  On the other hand, the proof of Theorem 8.4 uses entirely different techniques. In particular, it provides a characterization of the hash function optimized for a particular input.
>
> [The machine learned Oracle is assumed to be flawless at identifying the Heavy Hitters]
> Actually, this is not the case. The analysis in the paper already takes into account errors in the machine learning oracle. Please see the 2nd paragraph of Sec. 4.1 and Lemma 7.15. In summary, our results hold even if the learned oracle makes prediction errors with probability O(1/ln(n)). We will revise the text to make it clearer.

---

### Official Review · AnonReviewer2 · 2018-11-07
**Unclear problem setting**

**Rating:** 6
**Confidence:** 1

**Review:**

Quality/clarity:
- The problem setting description is neither formal nor intuitive which made it very hard for me to understand exactly the problem you are trying to solve. Starting with S and i: I guess S and i are both simply varying-length sequences in U.
- In general the intro should focus more on an intuitive (and/or formal) explanation of the problem setting, with some equations that explain the problem you want to work on. Right now it is too heavy on 'related work' (this is just my opinion).

Originality/Significance:
I have certainly never seen a ML-based paper on this topic. The idea of 'learning' prior information about the heavy hitters seems original.

Pros:
It seems like a creative and interesting place to use machine learning. the plots in Figure 5.2 seem promising.

Cons:
- The formalization in Paragraph 3 of the Intro is not very formal. I guess S and i are both simply varying-length sequences in U.
- In general the intro should focus more on an intuitive (and/or formal) explanation of the problem setting, with some equations that explain the problem you want to work on. Right now it is too heavy on 'related work' (this is just my opinion).

-In describing Eqn 3 there are some weird remarks, e.g. "N is the sum of all frequencies". Do you mean that N is the total number of available frequencies? i.e. should it be |D|? It's not clear to me that the sum of frequencies would be bounded if D is not discrete.
- Your F and \tilde{f} are introduced as infinite series. Maybe they should be {f1, f2,..., fN}, i.e. N queries, each of which you are trying to be estimate.
- In general, you have to introduce the notation much more carefully. Your audience should not be expected to be experts in hashing for this venue!! 'C[1,...,B]' is informal abusive notation. You should clearly state using both mathematical notation AND using sentences what each symbol means. My understanding is that that h:U->b, is a function from universe U to natural number b, where b is an element from the discrete set {1,...,B}, to be used as an index for vector C. The algorithm maintains this vector C\in N^B (ie C is a B-length vector of natural numbers). In other words, h is mapping a varying-length sequence from U to an *index* of the vector C (a.k.a: a bin). Thus C[b] denotes the b-th element/bin of C, and C[h(i)] denotes the h(i)-th element.
- Still it is unclear where 'fj' comes from. You need to state in words eg "C[b] contains the accumulation of all fj's such that h(j)=b; i.e. for each sequence j \in U, if the hash function h maps the sequence to bin b (ie $h(j)=b$), then we include the *corresponding frequency* in the sum."
- What I don't understand is how fj is dependent on h. When you say "at the end of the stream", you mean that given S, we are analyzing the frequency of a series of sequences {i_1,...,i_N}?
- Sorry, it's just confusing and I didn't really understand "Single Hash Function" from Sec 3.2 until I started typing this out.
- The term "sketch" is used in Algorithm1, like 10, before 'sketch' is defined!!
-I'm not going to trudge through the proofs, because I don't think this is self-contained (and I'm clearly not an expert in the area).

Conclusion:
Honestly, this paper is very difficult to follow. However to sum up the idea: you want to use deep learning techniques to learn some prior on the hash-estimation problem, in the form of a heavy-hitter oracle. It seems interesting and shows promising results, but the presentation has to be cleaned up for publication in a top ML venue.



******
Update after response:
The authors have provided improvements to the introduction of the problem setting, satisfying most of my complaints from before. I am raising my score accordingly, since the paper does present some novel results.

---

> ### Author Response · Authors · 2018-11-15
> **Response to Reviewer 2**
>
> Thank you for the thoughtful comments. We are glad that you found our algorithmic approach original, and our experiments promising.
>
> Regarding the notation, given that the topic of our paper is inherently interdisciplinary -- spanning machine learning and algorithm theory -- we need to use notions and notation from both communities. This can lead to misunderstandings, but there is no easy way around it. In the paper we tried to follow the notation used in heavy-hitter analysis in algorithm theory to make it easy to compare the analysis to past work. But since there is no standard notation across both fields, it is difficult to find a notation that is easily accessible to both communities.
>
> In addition, there are indeed a few places in the paper where our phrasing could have been better, thank you for pointing this out.  We discuss this in more detail below, and hope this should clarify any misunderstandings.
>
> Regarding our proofs, they are all self-contained.
>
> - The problem setting description is neither formal nor intuitive which made it very hard for me to understand exactly the problem you are trying to solve. Starting with S and i: I guess S and i are both simply varying-length sequences in U.
>
> To clarify, the input S is a sequence *of elements* from some universe U. To give an example, we could have U={0...65535}, in which case the sequence S would consist of integers in the range 0...65535. For example, S = 10101, 21222, 10222, 1, 10, 1, 52233, 62223 is an example sequence of length 8 whose items belong to U.
>
> The remainder of the problem definition is as described in the introduction: a frequency estimation algorithm reads the sequence S in one pass, and after that, for any element i from U, reports an estimate of  f_i,  the number of times element i occurs in S. In the above example, we have, e.g., f_1=2.
>
> - In general the intro should focus more on an intuitive (and/or formal) explanation of the problem setting, with some equations that explain the problem you want to work on. Right now it is too heavy on 'related work' (this is just my opinion).
>
> Thanks for the suggestions. We will include more explanation in the introduction and condense related work while keeping it thorough.
>
> - In describing Eqn 3 there are some weird remarks, e.g. "N is the sum of all frequencies". Do you mean that N is the total number of available frequencies? i.e. should it be |D|? It's not clear to me that the sum of frequencies would be bounded if D is not discrete.
>
>  N is the sum of all frequencies; i.e., N = \sum_{ i \in U }  f_i.
>
> - Your F and \tilde{f} are introduced as infinite series. Maybe they should be {f1, f2,..., fN}, i.e. N queries, each of which you are trying to be estimate.
>
> The series are indeed finite, we skipped the last index for simplicity. Formally, it should be F = {f_1, …, f_|U|} and ~F = {~f_1, …, ~f_|U|}
>
> - In general, you have to introduce the notation much more carefully. Your audience should not be expected to be experts in hashing for this venue!! 'C[1,...,B]' is informal abusive notation. You should clearly state using both mathematical notation AND using sentences what each symbol means.
>
> As stated, C[1...B] is a one-dimensional array. Equivalently, it is a B-dimensional vector. We refer to C as an “array” as opposed to “vector” for the sake of consistency with prior work on frequency estimation, and to avoid nested subscripts.
>
> C[b] indeed denotes the b-th element/bin of C. Regarding the notation h: U -> [B] : we use [B] to denote the set {1...B}. We define it in Section 7, but we should have defined it earlier. The formula h: U->[B] indeed denotes a function h that maps elements of U to {1...B}.
>
> - Still it is unclear where 'fj' comes from. You need to state in words eg "C[b] contains the accumulation of all fj's such that h(j)=b; i.e. for each sequence j \in U, if the hash function h maps the sequence to bin b (ie $h(j)=b$), then we include the *corresponding frequency* in the sum."
>
> We hope that after the earlier clarifications, the equation C[b] = sum_{j:h(j)=b} f_j  is more clear now.
>
> - What I don't understand is how fj is dependent on h. When you say "at the end of the stream", you mean that given S, we are analyzing the frequency of a series of sequences {i_1,...,i_N}?
>
> f_j does not depend on h, only on the input sequence S. Since an element j can occur anywhere in S, the equation C[b] = sum_{j:h(j)=b} f_j  holds only after the algorithm scans the whole sequence S.
>
> - The term "sketch" is used in Algorithm1, like 10, before 'sketch' is defined!!
>
> As explained in the description, items not stored in unique buckets “are fed to the remaining B − Br buckets using a conventional frequency estimation algorithm SketchAlg”. The word “sketch” in Algorithm 1 refers to the storage used by SketchAlg. To avoid confusion, we will shorten line 10 to “feed i to SketchAlg”.

---

### Official Review · AnonReviewer3 · 2018-11-08
**A good problem discussed and the proposed ML approach seems reasonable.**

**Rating:** 7
**Confidence:** 4

**Review:**

The authors are proposing an end-to-end learning-based framework that can be incorporated into all classical frequency estimation algorithms in order to learn the underlying nature of the data in terms of the frequency in data streaming settings and which does not require labeling. According to my understanding, the other classical streaming algorithms also do not require labeling but the novelty here I guess lie in learning the oracle (HH) which feels like a logical thing to do as such learning using neural networks worked well for many other problems.

The problem formulation and applications of this research are well explained and the paper is well written for readers to understand. The experiments show that the learning based approach performs better than their all unlearned versions.

But the only negative aspect is the basis competitor algorithms are very simple in nature without any form of learning and that are very old. So, I am not sure if there are any new machine learning based frequency estimation algorithms.

---

> ### Author Response · Authors · 2018-11-15
> **Response to Reviewer 3**
>
> Thank you for the thoughtful comments. We are glad that you found our problem interesting, and problem formulation/applications of this research well explained.
>
> Regarding the competing algorithms:  Both algorithms that we compare to, Count-Sketch and Count-Min, are state-of-the-art hashing-based algorithms (see e.g., Cormode & Hadjieleftheriou (2008)). Further, they are widely used in practice for processing internet traffic, large databases, query logs, web document repositories, etc.
>
> To the best of our knowledge, our paper is the first to use machine learning to design better sketches for any streaming problem. We tried to cover related work thoroughly in section 2.

---

### Author Response · Authors · 2018-11-22
**Minor updates to the paper**

Dear reviewers,

Thank you again for the thoughtful comments. We made minor updates in the paper (labeled in blue) to address some of the notation issues. We also included more explanation of our problem in the introduction. We hope that this helps clarify any misunderstandings. Please let us know if you have any other comments.

---

### Meta-Review · Area_Chair1 · 2018-12-13

**Confidence:** 5
**Recommendation:** Accept (Poster)

**Metareview:**

The paper conveys interesting ideas but reviewers are concern about an incremental nature of results, choice of comparators, and in general empirical and analytical novelty.